# Chaotic Dynamics are Intrinsic to Neural Network Training with SGD

**Luis M. Herrmann**
Center for Digital Health - AG Roland Eils
Berlin Institute of Health
Kapelle-Ufer 2, 10117 Berlin
`luis.herrmann@charite.de`

**Maximilian Granz**
FU BioRobotics Lab
Freie Universität Berlin
Arnimallee 7, 14195 Berlin
`maximilian.granz@fu-berlin.de`

**Tim Landgraf**
FU BioRobotics Lab
Freie Universität Berlin
Arnimallee 7, 14195 Berlin
`tim.landgraf@fu-berlin.de`

## Abstract

With the advent of deep learning over the last decade, a considerable amount of effort has gone into better understanding and enhancing Stochastic Gradient Descent so as to improve the performance and stability of artificial neural network training. Active research fields in this area include exploiting second order information of the loss landscape and improving the understanding of chaotic dynamics in optimization. This paper exploits the theoretical connection between the curvature of the loss landscape and chaotic dynamics in neural network training to propose a modified SGD ensuring non-chaotic training dynamics to study the importance thereof in NN training. Building on this, we present empirical evidence suggesting that the negative eigenspectrum - and thus directions of local chaos - cannot be removed from SGD without hurting training performance. Extending our empirical analysis to long-term chaos dynamics, we challenge the widespread understanding of convergence against a confined region in parameter space. Our results show that although chaotic network behavior is mostly confined to the initial training phase, models perturbed upon initialization do diverge at a slow pace even after reaching top training performance, and that their divergence can be modelled through a composition of a random walk and a linear divergence. The tools and insights developed as part of our work contribute to improving the understanding of neural network training dynamics and provide a basis for future improvements of optimization methods.

## 1   Introduction

In the last decade, the advent of Deep Learning has led to an explosive development of powerful machine learning models capable of solving ever more complex problems. However, while numerous different architectures have been developed, the underlying optimization procedure - Stochastic Gradient Descent (SGD) (Robbins, 2007) and its descendants (Duchi et al., 2011; Kingma and Ba, 2017) - has largely remained the same, with many aspects of its nature not being fully understood yet. The role of second order dynamics is one of those aspects, as are chaotic dynamics of Artificial Neural Network (ANN) training, both being notoriously difficult to investigate due to the high computational cost involved. In this paper, we establish a connection between the two aspects and generate new

36th Conference on Neural Information Processing Systems (NeurIPS 2022).

insights by studying the intersection of these two through a series of experiments conducted on small-sized models trained on natural datasets. Our main four contributions are as follows:

1. By modeling ANN training with SGD as a time-discrete dynamical system, we propose a modified SGD algorithm ensuring non-chaotic training dynamics to study the importance of chaos in ANN training.

2. We find empirical evidence suggesting that directions of negative curvature - and thus local chaos - cannot be removed without hurting the training performance of ANNs.

3. We show empirically that the network dynamics start out diverging exponentially at the beginning of the training but transition asymptotically against polynomial behaviour as the model performance converges.

4. Elaborating on the previous aspect, we show that even as the model training converges, the distance between similarly initialized models continues to grow at a small pace, and that this behaviour can be modelled as a sum of a linear divergence and a random walk.

## 2  Theoretical preliminaries

### 2.1  Definition of chaos

A popular saying characterizing the notion of chaos states that 'The flapping of a butterfly's wings can cause a storm on the other side of the world.', sometimes also briefly referenced as the butterfly effect. Formally, a dynamical system is chaotic if it is (1) **sensitive to initial conditions**, (2) topologically transitive and (3) has periodic points of the system which are dense in state space (Skokos, 2009, p.3, Def.1). Although a complete analysis of chaos dynamics of a dynamical system would encompass quantifying all three aforementioned properties, property (1) is the most relevant property for model optimization: Given a small deviation in the parameter state of a model at any point of the optimization, one should hope to obtain a similar solution at the end of the optimization procedure, and by extension a solution that performs similarly well both on training and validation data. Therefore, we use chaotic to refer to the sensitivity to initial conditions.[1]

### 2.2  Recap: Lyapunov Exponents

A common tool for characterizing chaos are Lyapunov exponents, which measure the evolution of displacement vectors over time. Consider a dynamical system $\boldsymbol{\theta}^{(t+1)} = \boldsymbol{f}(\boldsymbol{\theta}^{(t)})$, where $\boldsymbol{\theta}^{(t)}$ is the system state and $\boldsymbol{f}$ the dynamics function which determines how the system evolves over time. Given an initial, infinitesimally small displacement vector $\delta\boldsymbol{\theta}_0$ s.t. $\boldsymbol{\theta}'_0 = \boldsymbol{\theta}_0 + \delta\boldsymbol{\theta}_0$, the evolution of the displacement vector over time is given by the so-called tangent map, typically denoted $d_w\boldsymbol{\Phi}^{(t)}$ or $\boldsymbol{Y}^{(t)}$ in matrix notation, satisfying

$$\delta\boldsymbol{\theta}^{(t)} = \boldsymbol{Y}^{(t)}\delta\boldsymbol{\theta}^{(0)} = \frac{\partial\boldsymbol{\theta}^{(t)}}{\partial\boldsymbol{\theta}^{(0)}}\delta\boldsymbol{\theta}^{(0)}, \tag{1}$$

where the matrix $\boldsymbol{Y}^{(t)}$ is the Jacobian of the dynamics function at time $t$ w.r.t. the state at the initial time step. When the state $\boldsymbol{\theta}^{(t)}$ depends only on the state at the previous time point $\boldsymbol{\theta}^{(t-1)}$ the action of the tangent map can be decomposed through application of the chain rule to

$$\boldsymbol{Y}^{(t)} = \boldsymbol{J}_f(\boldsymbol{\theta}^{(t)})\boldsymbol{Y}^{(t-1)} = \frac{\partial\boldsymbol{\theta}^{(t)}}{\partial\boldsymbol{\theta}^{(t-1)}}\boldsymbol{Y}^{(t-1)}, \tag{2}$$

with $\boldsymbol{J}_f$ being the Jacobian of the dynamics function at time step $t$. The Lyapunov exponents are defined as the eigenvalues (the Lyapunov spectrum) of the matrix

$$\boldsymbol{\Lambda}^{(\infty)} = \lim_{t\to\infty} \frac{1}{2t} \ln \boldsymbol{Y}^{(t)T}\boldsymbol{Y}^{(t)}, \tag{3}$$

which we will refer to as *Lyapunov matrix* from now on. With this definition, we expect initial system perturbations $\Delta\boldsymbol{\theta}_0$ to evolve according to

$$\Delta\boldsymbol{\theta}(t) \propto e^{\lambda_1 t}\Delta\boldsymbol{\theta}_0, \tag{4}$$

---

[1]A more rigorous definition of chaos would require the system's invariant set to be bounded.

where $\lambda_1$ is the maximum eigenvalue of the Lyapunov matrix, also called **maximum Lyapunov Characteristic Exponent** (mLCE). The respective eigenvector gives us the direction of maximum expansion of an initial displacement as the dynamic system evolves. Thus, the mLCE can be used to characterize three different types of system dynamics:

1. $\lambda_1 > 0$: Nearby trajectories **diverge exponentially** (chaotic).
2. $\lambda_1 = 0$: Nearby trajectories **diverge polynomially** (edge-chaotic).
3. $\lambda_1 < 0$: The distance of nearby trajectories is **upper-bounded**.

Although the mLCE is sufficient to characterize the overall system dynamics, a more fine-grained analysis can be obtained in a natural manner by considering the full eigenvalue spectrum of the Lyapunov matrix, with the biggest eigenpair $(\lambda_1, \boldsymbol{v}_1)$ giving us the "most chaotic" direction of the system evolution, $(\lambda_2, \boldsymbol{v}_2)$ the second-most chaotic, and so on assuming $\lambda_1 \geq \lambda_2 \geq ... \geq \lambda_N$. Thus, for a small initial perturbation $\boldsymbol{\theta}'^{(0)} = \boldsymbol{\theta}^{(0)} + \varepsilon\boldsymbol{v}_i$, we expect the perturbation to evolve approximately as dictated by the respective eigenpair as $\Delta\boldsymbol{\theta}(t) \propto e^{\lambda_i t}\varepsilon\boldsymbol{v}_i$. Along any direction, we can expect the bound

$$e^{\lambda_N(t)t}\varepsilon \leq \Delta\boldsymbol{\theta}(t) \leq e^{\lambda_1(t)t}\varepsilon \tag{5}$$

to hold for sufficiently small $\varepsilon$ s.t. the first order approximation

$$\boldsymbol{\theta}'^{(t)} \approx \boldsymbol{\theta}^{(t)} + \boldsymbol{Y}^{(t)}(\varepsilon\boldsymbol{w}) \tag{6}$$

of the training process at time step $t$, where $\boldsymbol{w}$ is a unit-length perturbation axis, is accurate.

## 2.3 Curvature and chaos

A neural network evolving according to SGD with learning rate $\gamma$ (without momentum) and trained with data batches $\boldsymbol{z}^{(t)}$ at time step $t$ can be described by the equation

$$\boldsymbol{\theta}^{(t)} = \boldsymbol{\theta}^{(t-1)} - \gamma\boldsymbol{g}(\boldsymbol{\theta}^{(t-1)}; \boldsymbol{z}^{(t-1)}), \tag{7}$$

where $\boldsymbol{g}(\boldsymbol{\theta}^{(t-1)}; \boldsymbol{z}^{(t-1)})$ is the loss gradient w.r.t. to network parameters $\boldsymbol{\theta}^{(t-1)}$. We show in Appendix A.1 and A.3 that the evolution of the tangent map without and with momentum can be described as

$$\boldsymbol{Y}^{(t+1)} = (\boldsymbol{I} - \gamma\boldsymbol{H}^{(t)})\boldsymbol{Y}^{(t)} \quad \text{and} \quad \boldsymbol{Y}^{(t+1)} = \boldsymbol{Y}^{(t)} - \gamma\left(\sum_{s=1}^{t}\beta^{(t-s)}\boldsymbol{H}^{(s)}\boldsymbol{Y}^{(s)}\right), \tag{8}$$

respectively. This result shows that, discounting the effects of random batch sampling in SGD, the chaotic dynamics depend on the Hessian and thus on the curvature of the loss landscape during training.

## 2.4 Local chaos and negative Hessian eigenvalue spectrum

Given the connections we established above, one could hope to improve the training performance of SGD by using second-order curvature information to avoid/promote a chaotic evolution of the system, assuming chaos to be detrimental/beneficial for neural network training, e.g. by pruning the directions of maximum chaos from the gradient updates. Unfortunately, the Lyapunov matrix is not practical for this purpose because it considers the chaos over the entire training up to the current time point, so it can only inform us of a chaotic evolution of the system a posteriori. To mitigate this problem, one can instead consider a greedy approach where at any given time step $t$, one looks at the finite-time Lyapunov matrix after a single time step, i.e.

$$\boldsymbol{\Lambda}^{(t,t+1)} = \frac{1}{2}\ln(\boldsymbol{I} - \gamma\boldsymbol{H})^T(\boldsymbol{I} - \gamma\boldsymbol{H}) = \frac{1}{2}\ln(\boldsymbol{I} - \gamma\boldsymbol{H})^2 \tag{9}$$

The eigenvalues of this matrix are called the **Local Lyapunov Exponents (LLE)** at time step t. Analogously to the LCEs, we say the system is *locally chaotic* in the direction of eigenpair $(\lambda_i, \boldsymbol{v}_i)$ iff $\lambda_i > 0$. Using the fact that $\boldsymbol{\Lambda}^{(t,t+1)}$ has the same eigenvectors as $\boldsymbol{H}$, we can relate the eigenspectrum of the Hessian to local chaotic dynamics as follows:

**Theorem 2.1.** *Given a neural network with N parameters, let $\boldsymbol{H}$ be the Hessian of the loss w.r.t. network parameters, and let $(\lambda_i, \boldsymbol{v}_i)_{i=1}^N$ be the Hessian's eigenpairs. Furthermore, let us assume that the network is trained using SGD with learning rate $\gamma$ and no momentum. Then, the network's LLEs indicate locally chaotic training behaviour in the direction of $\boldsymbol{v}_i$ if*

$$\lambda_i < 0 \lor \lambda_i > \frac{2}{\gamma} \tag{10}$$

For a proof see Appendix A.5. Our result leads to the interesting observation that sufficiently large positive eigenvalues (as determined by the learning rate) imply locally chaotic training behaviour in SGD. More interestingly however, negative eigenvalues automatically lead to locally chaotic training behaviour.

## 2.5 Pruning chaotic updates

Since the eigenpairs $(\lambda_i, \boldsymbol{v}_i)_{i=1}^N$ of $\boldsymbol{\Lambda}^{(t,t+1)}$ give us the directions of local chaos ordered by magnitude of the LLEs, we can remove the locally chaotic directions from the parameter updates in SGD by projecting the update vector onto non-locally-chaotic eigenvectors of $\boldsymbol{\Lambda}^{(t,t+1)}$. More specifically:

**Theorem 2.2.** *Given a neural network with Hessian $\boldsymbol{H}^{(t)}$, eigenpairs $(\lambda_i^{(t)}, \boldsymbol{v}_i^{(t)})_{i=1}^N$ of $\boldsymbol{\Lambda}^{(t,t+1)}$ and gradient updates $\Delta\boldsymbol{\theta}^{(t)}$, let there be $k \leq N$ chaotic eigenpairs, i.e. $\lambda_1 \geq ...\lambda_k > 0$. Suppose all chaotic components are removed through projection, leading to a modified update vector*

$$\Delta\tilde{\boldsymbol{\theta}}^{(t)} = \boldsymbol{V}^{(t)}(\boldsymbol{V}^{(t)})^T \Delta\boldsymbol{\theta}^{(t)}, \qquad \boldsymbol{V}^{(t)} = \sum_{l>k} \boldsymbol{v}_l^{(t)} \boldsymbol{v}_l^{(t)^T}. \tag{11}$$

*Then, for training with SGD without momentum, the updated training dynamics are given by*

$$\boldsymbol{\theta}^{(t+1)} = \boldsymbol{\theta}^{(t)} - \gamma\boldsymbol{V}^{(t)}\boldsymbol{V}^{(t)^T}\boldsymbol{g}(\boldsymbol{\theta}^{(t)}), \qquad \boldsymbol{Y}^{(t+1)} = (\boldsymbol{I} - \gamma\boldsymbol{V}^{(t)}\boldsymbol{V}^{(t)^T})\boldsymbol{Y}^{(t)} \tag{12}$$

*and for sufficiently small variations of the initial parameters, the system is guaranteed not to have exponentially diverging orbits.*

Proof is provided in Appendix A.7.

# 3 Related Work

Since the full Lyapunov eigenspectrum is very expensive to calculate, previous work analyzing chaotic dynamics has so far mostly been limited either to small networks or to the largest mLCE, as investigated for instance by Das et al. (2000). Several investigations of chaos in feedforward ANNs such as those by Li (2019); Feng et al. (2019); Zhang et al. (2021) have focused on sensitivity of outputs w.r.t. the inputs rather than w.r.t. the training dynamics, modelling the network function itself as a discrete dynamical system. Recently, Vogt et al. (2020) have calculated approximations of the Lyapunov exponents in RNNs and found correlations between better generalization (through lower validation loss) and a smaller mLCE. To our knowledge, Sasdelli et al. (2021) are the only ones to have investigated chaos dynamics in ANN training by SGD, but interestingly, they model ANN training by SGD as an approximately time-continuous dynamical system and not more accurately as a time-discrete dynamical system, thus arriving at slightly different theoretical results. While their theoretical work also links the biggest Lyapunov exponent to the most negative eigenpair of the network Hessian, positive eigenpairs as indicators of chaos disappear in their considerations, which is not the case for time-discrete SGD without further assumptions. Unfortunately, they do not investigate how the directions of chaotic evolution in general and negative curvature in particular affect the training process.

There is a considerable body of work on the eigenvalue spectra of ANN Hessians that can be tied into our theoretical findings, including an analysis of the full eigenvalue spectrum of a small 784-2-10 MLP for MNIST digit classification by Sagun et al. (2017). Among other findings, the authors remark a two-phase distribution of eigenvalues with a bulk centered around zero and large positive top eigenvalues (although not sufficiently big to satisfy $\lambda > 2/\gamma$), as well as the **presence of negative eigenvalues throughout the entire training**. Further analysis of the negative spectrum by Alain et al. (2019) has indicated that individual negative eigenvalues on average lead to the biggest improvement

of the loss in neural networks, but that current optimization algorithms appear not to be good at optimizing in directions of negative curvature. This finding was extended by Gur-Ari et al. (2018), who found that the gradient of the Hessian mostly resides in the space spanned by the top Hessian eigenvectors and does not mix with the bulk, suggesting a marginal importance of the negative Hessian eigenpairs for the training dynamics.

To the best of our knowledge, we are the first to use time-discrete modelling of training dynamics by SGD to derive predictions of how curvature affects chaotic dynamics, and the first to explore extensively - using the full Hessian spectrum at every training step - how directions of different curvature affect the dynamics of the training process. This extensive investigation provides us with empirical evidence allowing us to establish a novel connection between chaotic training dynamics and the training performance.

## 4 Experiments

Taking into account all aspects discussed so far, we can make the following claims:

1. Due to the presence of negative eigenvalues in all training steps and assuming the MLP for MNIST is a representative model for Feedforward ANNs, **ANNs can be expected to have positive LLEs at every training step**.

2. Unless the positive top eigenvalues of the Hessian are sufficiently big to satisfy $\lambda > 2/\gamma$ - which was not the case for the MLP for MNIST investigated by Sagun et al. (2017) - the **chaotic behaviour of ANN training by SGD is exclusively determined by negative eigenvalues in the bulk**.

What remains to be verified experimentally is whether the negative eigenspectrum has little or at worst a detrimental effect on the training performance, as suggested by the findings of Alain et al. (2019) and Gur-Ari et al. (2018), or if it is actually benificial to the performance of ANN training.

**Methods**   The code for all our experiments is written in Python using the Pytorch framework (Paszke et al., 2017), and available at GitHub [2]. For the calculation of the Hessians, we use O2Grad (Anonymous, 2022), a package on top of Pytorch that enables faster calculation of the Hessian of small ANNs using 2nd order backpropagation. As datasets, we use USPS (Hull, 1994) and FashionMNIST (Xiao et al., 2017) (with images subsampled to $16 \times 16$ pixels), since these datasets contain sufficiently low-dimensional, natural data to allow for the calculation of the Hessian and its eigenvalue decomposition at every training step, both for a 784-20-10 MLP and for a small 2D CNN. To modify the parameter updates of the model as proposed in Theorem 2.2, we use the Pytorch implementation of SGD and alter it slightly to implement an own class CGD (Chaos-sensitive Gradient Descent) with the ability to filter parts of the eigenvalue spectrum.

Our experiments were run on single GPU nodes of a system featuring an AMD Ryzen Threadripper 1950X processor, 4x Nvidia GeForce RTX 2080 TI GPUs (11GB VRAM) and 64GB RAM, as well as on a second system featuring an Intel(R) Core(TM) i5-8600K, Nvidia GeForce GTX 1080 Ti (11GB VRAM) and an Nvidia Titan XP (12GB VRAM) and 32GB of RAM. Both systems run on Debian Debian GNU/Linux 11 (bullseye).

### 4.1   Local Chaos Investigation

**Chaotic direction pruning deteriorates training performance**   In order to verify the impact of pruning different components on the training performance, we equipped CGD with the ability to filter (1.) chaotic eigenpairs, (2.) only negative eigenpairs and (3.) only positive eigenpairs, and out of those, filter only the $k$ largest by absolute value or prune $k$ at random. As our results for the MLP on USPS in Figure 1 show, models trained without pruning (i.e. regular SGD) reach high/low training accuracies/losses, while the models trained with CGD and pruning of chaotic eigenvalues get stuck at much lower/higher training accuracies/losses. The same phenomenon can be observed for validation losses/accuracies, and on the 2D CNN and FashionMNIST (see Appendix B.1). This is remarkable because it seems to suggest that locally chaotic training behaviour is essential for ANN training in order to quickly achieve reasonable training performance and generalization, and that ANNs trained

---

[2] https://github.com/luisherrmann/chaotic_neurips22

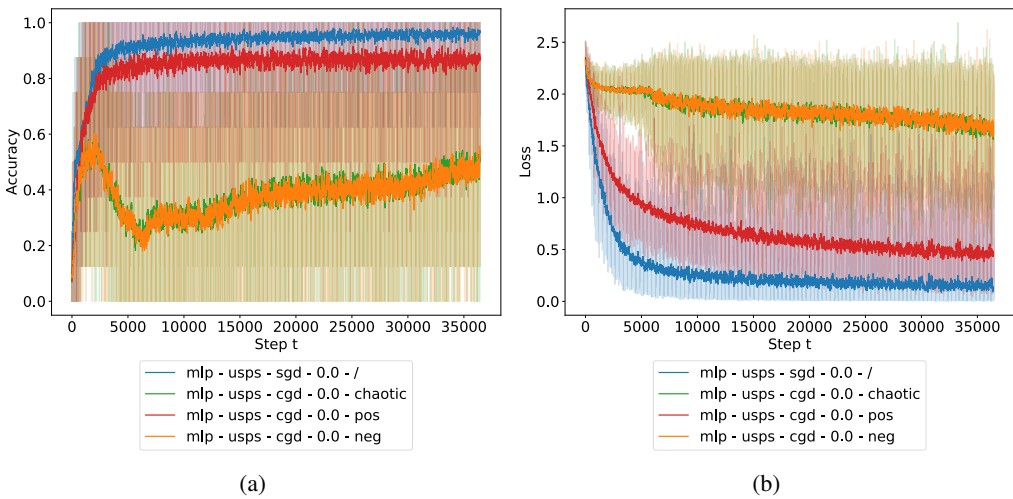

Figure 1: Line plots of the (a) accuracy and (b) loss curves for an MLP (relu activation) trained on the USPS dataset without momentum. The models trained with CGD use pruning of the full chaotic, negative or positive Hessian spectrum at every time step. The smoothed, solid lines are obtained through local averaging (window size 50) of the respective sequential data. Pruning of the full positive spectrum (red) only has a small effect on training performance, while negative and chaotic pruning strongly limits the network's ability to learn.

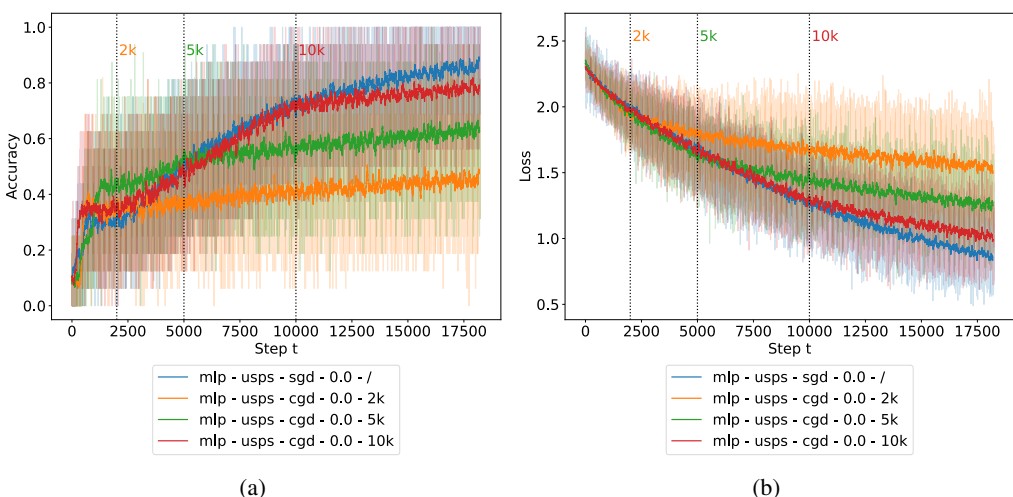

Figure 2: Line plots of the (a) accuracy and (b) loss curves for an MLP trained on the USPS dataset without momentum. We use a sigmoid activation because this slows down convergence and makes the splitting of performance metrics better visible. The models trained with CGD use pruning of the full chaotic Hessian spectrum at every time step after the onset of pruning after 2k, 5k and 10k time steps. The smoothed, solid lines are obtained through local averaging (window size 50) of the respective sequential data). The accuracy and loss curves start diverging from the baseline curve when pruning starts.

by SGD are hardly able to learn in the absence of locally chaotic directions. To make sure this is not an effect occurring only in the early stages of training by SGD, we execute several runs where the pruning of locally chaotic directions starts later in the training. As we can see in Figure 2, training loss and accuracy start improving at a slower rate after the onset of pruning, suggesting that the models learn far more slowly as soon as the pruning of chaotic directions is engaged.

**Negative Hessian eigenspectrum drives training by SGD**  Interestingly, the number of positive chaotic eigenvalues of the Hessian is effectively 0 throughout most of the training (see Appendix

B.2). This means that the chaotic directions pruned during the training come almost exclusively from the negative eigenpairs of the Hessian, which is in line with our previously formulated expectation. However, the extent to which pruning of the negative eigenvalues hinders the training of the network suggests that the negative eigenspectrum of the Hessian is far more important to the training dynamics of neural networks than previously believed by Alain et al. (2019). To better understand the role of the negative and positive eigenspectrum of the Hessian, we perform several runs where we prune (1.) the **entire negative eigenspectrum** and (2.) **the entire positive eigenspectrum** of the Hessian.

As expected, our results in Figure 1 for pruning the full negative eigenspectrum coincide with the results for pruning the full chaotic eigenspectrum of the Hessian: Models trained with pruning of the full negative eigenspectrum of the Hessian reach far lower training performance than models trained with regular SGD. However, **pruning the full positive eigenspectrum of the Hessian only has a small impact on the training performance**: Although the accuracy/loss reaches slightly lower/higher levels than for SGD without pruning, the ANNs trained in this way fare much better than their counterparts with negative pruning. We observe this behaviour across different datasets and models (see Appendix B.1).

These empirical results lead us to the following conclusions:

1. **The negative eigenvalue spectrum of the Hessian contains important information** about the optimization problem since it cannot be discarded without hurting training performance.

2. **The positive eigenvalue spectrum of the Hessian contains less important information** about the optimization since discarding it has less impact on the training.

3. Since the negative eigenspectrum induces locally chaotic training dynamics, **locally chaotic training dynamics are inevitable in SGD** without hurting training performance.

These appear to contradict the findings of Gur-Ari et al. (2018), who observe that the overlap of the gradient is biggest with the positive eigenpairs of the Hessian - an observation that we confirm in our experiments (see Appendix B.2). The apparent contradiction can be resolved through a simplified model of ANN training by SGD, in which **negative curvature precedes positive curvature**: Directions of negative curvature indicate high potential for future loss improvement from moving along those axes, while directions of positive curvature indicate proximity to a local minimum along said axes, and thus reduced potential for loss improvement (see Appendix B.3.1). **Removing the negative components** of the Hessian from the gradient **eliminates possibilities for big future improvements of the loss** and may only allow the optimizer to finish converging in the directions where it was moving anyway, hence why the network is still able to decrease its loss even when the negative eigenspectrum is pruned, though at a smaller pace. This also matches our observation that the count of negative/positive Hessian eigenvalues decreases/increases as the training proceeds (see Appendix B.2).

Note that this model does not explain why the gradient overlap with axes of positive curvature is greater than with axes of negative curvature, only why the former and a high importance of the negative eigenvalues can exist at the same time: Negative curvature is important during an initial discovery phase, and positive curvature during a subsequent exploitation phase.

### 4.2 Global Chaos Investigation

Our experiments so far have relied on a local quantification of chaos via the LLEs. Next, we extend our investigations to global chaos using finite-time Lyapunov exponents to quantify chaos over longer time spans.

**Finite-time Lyapunov exponents** In order to better understand long-term chaotic dynamics, we calculate the tangent maps from equation (8) at every time step $t$ and apply eigenvalue decomposition on the resulting finite-time Lyapunov matrix $\mathbf{\Lambda}^{(t)}$ to end up with the corresponding finite-time Lyapunov eigenspectrum. The Lyapunov exponents are above 0 at the beginning of the training and converge against 0 as training progresses (see C.1), suggesting the evolution is initially chaotic, but edge-chaotic in the time limits. Note that since the exponents do not drop below 0, we should expect models with similar initializations to continue diverging even after they reach top performance.

**Perturbation analysis**  The divergence behavior can be further explored by utilizing the Lyapunov eigenspace to perturb models along determined directions. As discussed in section 2.2, this space describes the global divergence directions locally around an initial set of parameters. In our perturbation experiments, we train a model A until convergence at time $T_{conv}$ and calculate the Lyapunov eigenpairs. We then use those eigenpairs to apply initial perturbations onto several models $B_1, ..., B_n$ initialized with the same random seed as A, and observe how their parameters evolve compared to A along the perturbation axes. The latter are derived from the finite-time subspaces

$$V_{\text{chaotic}}(t) := \{\boldsymbol{v} \in \text{Eig}(\boldsymbol{\Lambda}^{(t)}, \lambda) \mid \lambda > 0\}, \qquad V_{\text{non-chaotic}}(t) := \{\boldsymbol{v} \in \text{Eig}(\boldsymbol{\Lambda}^{(t)}, \lambda) \mid \lambda < 0\}$$

according to 5 perturbation strategies: We take (1) the top eigenvector of $\boldsymbol{\Lambda}^{(N)}$ as the **maximally chaotic** direction and (2) the bottom eigenvector as **maximally non-chaotic** direction. Furthermore we consider a linear combination of vectors sampled randomly from (3) $V_{\text{chaotic}}(N)$ and (4) $V_{\text{non-chaotic}}(N)$ that describe random **chaotic** and random **non-chaotic** directions, respectively. Finally, we also sample (5) a completely **random** direction for comparison. The initial parameters are perturbed along the respective directions, where we test different magnitudes $\varepsilon_1, \ldots \varepsilon_k$ for each direction to make sure we consider perturbation magnitudes that are in agreement with the locality required by our theoretical preliminaries (see equation (6)).

Ghorbani et al. (2019) finds that the loss landscape near a local minimum is almost flat ($\boldsymbol{H} \approx 0$) and thus using the finite-time Lyapunov matrix at a time point $t = T_{conv}$ where the training is sufficiently converged should provide an acceptable approximation. Therefore, the tangent map $\boldsymbol{Y}^{(t)}$ at the next time step $t > T_{conv} + 1$ and its eigenspace would barely change.

$$\boldsymbol{Y}^{(T_{conv}+1)} = (\boldsymbol{I} - \gamma \boldsymbol{H}^{(T_{conv})})\boldsymbol{Y}^{(T_{conv})} \approx \boldsymbol{Y}^{(T_{conv})} \tag{13}$$

Although we also see a similar distribution of the Hessian eigenspectrum, we do not observe a full convergence of the Hessian to zero (see Appendix B.2). However, the subspaces of chaotic and non-chaotic eigenvectors do stabilise towards the end of the training (see Appendix C.2), making a finite-time analysis with the above perturbation strategies sound.

**Distance results**  Figure 3 (a) shows the distance evolution of an MLP without momentum on USPS and FashionMNIST. For a comparison between model architectures and analysis of momentum, see Appendix C.3. In general, the maximal divergence (empirically $\approx 1$ in Figure 3 (a), up to 12 in further experiments in Appendix C.3) lies in range of the expected distance of standard initialized models in Pytorch (LeCun et al., 2012) for the duration of our experiments:

$$\mathbb{E}_{\boldsymbol{\theta}_{1,i}, \boldsymbol{\theta}_{2,i} \sim \mathcal{U}(-\frac{1}{\sqrt{k_i}}, \frac{1}{\sqrt{k_i}})}(\|\boldsymbol{\theta}_1 - \boldsymbol{\theta}_2\|_2) \approx 6.38,$$

where $k_i$ are the number of input features of layer $i$ (see Appendix A.8). Still, a surprising result is that the **distances do not seem to saturate** for the duration of our experiments, **which would be expected for a valley of finite size**. In fact, we are still reaching a similar performance with perturbations of magnitudes up to $\varepsilon \leq 7.5$, and only for $\geq 10$ there seems to be a significant drop in performance (see Appendix C.3).

Theoretically, the observed distance evolution can be explained as composition of linear divergence and a random walk. Suppose the gradients $\boldsymbol{g}_1(t), \boldsymbol{g}_2(t)$ have distance components $\boldsymbol{w}_1(t), \boldsymbol{w}_2(t)$ with constant distance $\alpha := \|\boldsymbol{w}_1(t) - \boldsymbol{w}_2(t)\|_2^2$

$$\boldsymbol{g}_i(t) \approx \boldsymbol{w}_i(t) + \boldsymbol{\eta}(t) \tag{14}$$

and noise components defined by $\boldsymbol{\eta}_1(t), \boldsymbol{\eta}_2(t) \in \mathcal{N}(0, \sigma^2)$. Then the distance $\Delta\theta(t)$ between two sets of parameters $\boldsymbol{\theta}_1^{(t)}$ and $\boldsymbol{\theta}_2^{(t)}$ evolves as follows:

$$\Delta\theta(t) \approx \mathbb{E}\left[\left\|\sum_{s=0}^t (\boldsymbol{w}_1(s) - \boldsymbol{w}_2(s)) + \sum_{s=0}^t (\boldsymbol{\eta}_1(s) - \boldsymbol{\eta}_2(s))\right\|_2\right] = \sqrt{\alpha t^2 + \beta t}, \tag{15}$$

with $\beta := 2D\sigma^2$ (see Appendix A.10). For smaller $t$, this term is still influenced by the random walk term $\beta t$ whereas for large $t$ the $\alpha t^2$ term dominates and leads to linear divergence. We give examples of fitting this baseline to the data in Figure 3(b).

Unfortunately, we were unable to explore whether the model distances saturate for a sufficiently long time evolution, which should be the case if the models end up in a bounded loss valley, whereas an

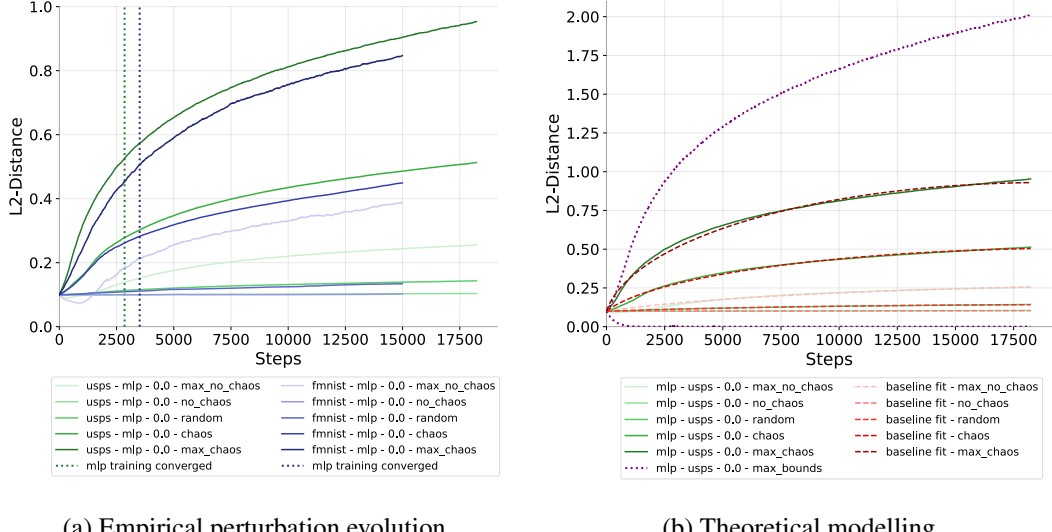

(a) Empirical perturbation evolution  (b) Theoretical modelling

Figure 3: Euclidean distance between a baseline MLP and a perturbed MLP trained with SGD (without momentum). We use 5 different strategies for computing initial perturbations of norm $\varepsilon = 0.1$ using the Lyapunov eigenspace of a model trained beyond convergence of training accuracy. Darker lines represent perturbation directions which we expect to lead to higher model divergence. The observed divergence in (a) on USPS (green) and on FashionMNIST (blue) increases even after convergence of the training accuracy across all perturbation directions. As shown in (b) for the USPS curves, the divergence can be approximated through a function $\sqrt{\alpha t^2 + \beta t}$ (red) within the observed time range. The choice of $\varepsilon$ for the trajectories is in agreement with the theoretical bounds (purple) predicted by the maximum/minimum LCE $\lambda_1(t)/\lambda_N(t)$ calculated at every time step.

unbounded loss valley would allow for model divergence to be unbounded as well. A more conclusive analysis on the matter will have to be provided by further studies. In any case, the evolution of training dynamics from chaotic to edge-chaotic in the parameter domain is interesting given that ANNs transition from unchaotic to chaotic in the input domain (Feng et al., 2019), suggesting there may be a complementary relation between the two.

**Choice of perturbation length** We present the experiments with a perturbation length of $\varepsilon = 0.1$ as we do not find that the approximation holds for larger $\varepsilon$. Surprisingly, we see a similar behavior for smaller perturbation lengths $\varepsilon < 0.1$: the smaller the differences get, the more randomly they behave. We argue that this phenomenon is caused by noise, probably induced by a numerical error. For further analysis see Appendix C.4.

**Theoretical bounds** In Figure 3 (b), we present the theoretical bounds derived from the maximal and minimal LCE (see equation (5)). Both bounds hold for our choice of $\varepsilon = 0.1$ over the entirety of our experiments where the lower bound converges always to zero (see Appendix C.5). The upper bound, on the other hand, can be utilized as a measure of maximal divergence of a single training run without conducting the full scale of our experiment. Note that for this purpose, it suffices to calculate a Lyapunov-vector-product which can be realized through a Hessian-vector-product for which faster algorithms exist (Pearlmutter, 1994).

# 5 Discussion

Although we have done our best to perform the pruning experiments on common datasets and model architectures, the experiments are currently difficult to extend to higher-dimensional datasets and models due to the high expense of calculating the Hessian at every training step. For similar reasons, we have not studied the effect of BatchNorm (Ioffe and Szegedy, 2015) and skip connections (He et al., 2015) yet, although we intend to do this at a later time. The high computational cost also limits training time used in our analysis and it could be that some aspects don't apply for longer

training times, e.g. our model for perturbed network divergence (see 3 (b)). In this context, it would also be interesting to investigate possible connections between the edge-chaotic dynamics later in training and overfitting, but we mainly limited the scope of our analysis to training metrics. Another aspect requiring further inquiry is why the model divergence is not always ordered according to our expectations for different perturbation strategies.

In this paper, we have shown that chaotic dynamics of ANN training by SGD are linked to negative curvature. We have found strong evidence that using negative curvature information is essential for training, and we have concluded that chaotic behaviour is inherent to training by SGD. Although globally chaotic dynamics are mostly present at the beginning of the training behaviour and fade towards the end of training, we have shown that models with slightly different initializations continue to diverge linearly in parameter space and provided a model for this behavior. Our work provides a theoretical starting point and tools for further investigations aiming to advance the understanding of chaos in ANN training. In this context, we especially recommend investigating the divergence of trajectories at the end of the training and exploring possible connections to generalization capability (see A.1), as well as the relation between chaos in the input and in the parameter domain.

## 6   Acknowledgements

We are deeply thankful to Julius Upmeier zu Belzen and Roland Eils from the Berlin Institute of Health for providing the main author with the time and support to write the paper. We would also like to thank Leon Sixt from the Biorobotics Lab and Martin Aleksandrov from the Dahlem Center for Machine Learning and Robotics for giving us valuable advice for the preparation and rebuttal of the paper. Maximilian Granz was supported by the Elsa-Neumann-Scholarship by the state of Berlin. We are also grateful to Nvidia for providing us with a Titan Xp and to ZEDAT for granting us access to their HPC system. Finally, we would like to thank the anonymous reviewers of our paper for taking their time to review our work and for supplying interesting points of discussion.

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
