# OpenReview forum: "Chaotic Dynamics are Intrinsic to Neural Network Training with SGD"
_NeurIPS.cc/2022/Conference — NeurIPS 2022 Accept_

### Official Review · Reviewer_LtWX · 2022-07-09

**Rating:** 6
**Confidence:** 4
**Soundness:** 3 good
**Presentation:** 3 good
**Contribution:** 3 good

**Summary:**

This manuscript mainly investigates a theoretical link between the curvature of the loss function and (local) chaotic dynamics in neural networks training through demonstrating the negative eigenvalue spectrum of the Hessian. Their empirical evidence verifies that the negative eigenspectrum and so the directions of local chaos cannot be removed from SGD without hurting training performance.

**Questions:**

Some effects of the local chaotic dynamics on the training performance is discussed through some experiments. However, I would like to see:  can the relation between local and long-term chaos be proven (mathematically) to demonstrate some problematic effects of local chaos on the training theoretically as well?


**Limitations:**

The authors have discussed the limitations of the work and there is no potential negative societal impact.


**Strengths And Weaknesses:**

**Strengths**

The main idea of the paper is interesting and it is nice to see a paper that is theoretically principled, and goes about formalizing some dynamical behavior arising in neural networks training.

**Weaknesses**

 I checked most of the proofs in detail and spot some errors; for instance:

1) It seems that Theorem 2.1 cannot hold in a general case (as is), and some assumptions on the Hessian of the loss ($H$) are required to be considered. More clear, in the proof of this theorem (Appendix A.5, line 522) it is mentioned that "Since all eigenvalues of $(I − \gamma H)^2$ are positive, the matrix is positive definite". But, all eigenvalues of $(I − \gamma H)^2$ are not necessarily positive; e.g. when $\gamma H$ has some eigenvalues equal to $1$. I mean let $H $ be a matrix such that  $\det(I − \gamma H) = 0$, then $(I − \gamma H)^2$ can have zero eigenvalues. For example, let $H= diag(\frac{1}{\gamma}, b, c)$, then $(I − \gamma H)^2 $ obviously has at least one zero eigenvalue. This affects the rest of the proof.

2)  Appendix A.1: In Eq. (19), from right to left, the second term should be "$\frac{\partial \theta_i^{(t)}}{\partial \theta_j^{(t)}}$".

3) Appendix A.2: It seems that Eq. (22) is written for $t=0$? So it needs to be modified for $t=1$. Also is there the assumption $v(0)=0$ for obtaining Eq. (22)?

There are also some minor issues and typos:

- The definition of chaotic systems (line 47) is for "smooth" dynamical systems (DS), but (it is well known that) it is no longer true for "non-smooth" DS such as RNNs with ReLU activation function (as piecewise linear DS). Especially, since piecewise linear DS can observe a specific kind of chaos called "Robust Chaos" in the absence of periodic windows and coexisting attractors; please see for instance the paper by Banerjee, Yorke and Grebogi: https://doi.org/10.1103/PhysRevLett.80.3049.

- Positive mLCE is a standard signature of chaos provided that the system's invariant set is **bounded** (as in this case two nearby orbits separate exponentially fast, but at the same time their mutual separation cannot go to infinity so that there are also folds). Otherwise a positive mLCE could be related to a diverging orbit which is not chaotic. Therefore, the authors should consider the assumption of "boundedness" and perhaps add it to Section 2.2.

- Is the reference " Robbins, H. E. (2007). A stochastic approximation method. Annals of Mathematical Statistics, 22:400–407" (line 369) correct? I couldn't find such a reference.

- Line 89, A.3 ---> A.2

---

> ### Author Response · Authors · 2022-08-02
> **Response to Reviewer LtWX**
>
> Thank you for your detailed review, we are glad you appreciated our theoretically principled approach. We have corrected most of the issues pointed out by you in our manuscript. Please let us know if you agree with the way we approached said issues:
>
> - You are right that the eigenvalues of $(I - \gamma H)$ are generally positive or zero, making the matrix positive-semidefinite. However, the kernel eigenvectors are not interesting from a dynamics perspective, since they would correspond to directions of immediate contraction, or more formally: If $(1/\gamma, \delta v)$ is an eigenpair of the Hessian, this implies $\delta v^{(t+1)} = J_f^{(t+1)}\delta v^{(t)} = 0$, so the parameters of the network don't change along this axis. We have extended theorem 2.1 to deal with this edge case.
> - Corrected.
> - We assume $v(0) = 0$. This is in line with how non-Nesterov momentum works in [Pytorch](https://pytorch.org/docs/stable/generated/torch.optim.SGD.html).
> - Thank you for pointing this out, we have accounted for this by correcting the definition of chaos to be for smooth dynamical systems.
> - This is a fair point of criticism that was also pointed out by reviewer 1. Strictly speaking, sensitivity to initial conditions (as characterized by the mLCE) is a necessary, but not sufficient condition for chaos and we should also require boundedness to describe the system dynamics as properly chaotic.
> However, we think the property of boundedness would be meaningful in a dynamical system with a long-term dynamical evolution, where orbits diverge exponentially but fold back on themselves in the bounded invariant set, implying the trajectories are unpredictable or “erratic”, in a sense. However, in the context of neural network evolution we think it is more interesting to know whether the trajectories are “erratic” in the early stages of training, i.e. in a time-local context, justifying our investigations on local chaos via local Lyapunov exponents.
> Our results are inconclusive on the question whether it would be justified to speak of global chaos in the canonical sense as time goes to infinity. The Lyapunov exponents appear to converge against 0, suggesting edge-chaotic behavior, but the divergence of the trajectories never fully saturates even for very small perturbations in the time-frames investigated. As we see it, this could mean two things:
>     1. The distances do eventually saturate, in which case the invariant sets should be bounded and the conventional definition of edge-chaotic would apply to the late stage of training.
>     2. The distances do not saturate, in which case there is no bounded invariant set and the trajectories just end up moving apart along some axes mixed with noise, as proposed in our model. In this case, this phase of the training could not be described as edge-chaotic, but the finding would be intriguing because it would mean that NNs trained by SGD do not converge against a finite region in parameter space.
>
>     To avoid confusion and an overloading of the word “chaotic”, we have tried to carve out more clearly that our claims on chaotic dynamics refer to the locally chaotic dynamics visible in the early stages of the training, but are more uncertain for the long-term dynamics of the network. We have adapted the manuscript accordingly and would value your feedback.
>
>
> - We have corrected the year of the reference to 1951.
> - Corrected.
>
> ## Questions:
>
> If we understand your question correctly, you are contemplating the possibility that there could be a tradeoff between the apparent benefits of local chaos (by virtue of the negative Hessian eigenvalues) and some negative impact mediated through long-term chaos or sensitivity to initial conditions, e.g. worse generalization properties.
>
> Our current empirical results do not suggest worse generalization properties, although we might have to investigate this more extensively. However, we have provided a mathematical argument at the end of the proof section in the supplementary material for why sensitivity to initial conditions could in theory affect generalization properties.

---

> > ### Comment · Reviewer_LtWX · 2022-08-08
> > **Thanks for your response!**
> >
> > Thank you very much for your detailed response!
> > I also appreciate you extended Theorem 2.1 as well as the additional clarifications. I think Theorem 2.1 seems to be correct now. Also, most of my concerns have been addressed and I have revised my score upward.

---

### Official Review · Reviewer_tUai · 2022-07-11

**Rating:** 3
**Confidence:** 3
**Soundness:** 2 fair
**Presentation:** 2 fair
**Contribution:** 2 fair

**Summary:**

This paper studies neural network training dynamics via the lens of the Hessian eigenspectrum. In particular, the authors show that the chaotic dynamics in neural network training are largely determined by the negative eigenvalue of the Hessian. Furthermore, the authors empirically demonstrate that the negative eigenvalue direction is important for making progress in training.

**Questions:**

See the weaknesses mentioned above.

Other questions and suggestions:
- I suggest the authors clarify the main advantages of Lyapunov exponent analysis over existing analyses. If I understand correctly, most results derived in section 2 can be obtained in a toy quadratic model.
- To scale up to larger datasets, the authors could check out literature about the Lanczos algorithm and its stochastic version (see e.g. [1]).



References:
1. An Investigation into Neural Net Optimization via Hessian Eigenvalue Density

**Limitations:**

The authors discussed the limitations at the end of the paper. I don't see any negative societal impact of this work as it is an "understanding" paper.

**Strengths And Weaknesses:**

Strengths:
- The discussion about the role of negative eigenspectrum in optimization is interesting.

Weaknesses:
- The theoretical model of Lyapunov exponent offers no further insights into the training dynamics compared to standard quadratic models or linearized models.
- The fact that negative eigenvalues would lead to chaotic dynamics is well known in the optimization community. Basically, I feel the whole section 2 is just background and I don't really know what's the contribution of this work.
- All experiments are done in very toy settings. The authors only conducted experiments on FashionMNIST and USPS, which I don't think very convincing.

---

> ### Author Response · Authors · 2022-08-02
> **Response to Reviewer tUai**
>
> Thank you very much for taking your time to review our paper. We are happy to learn you found our investigations on the contribution of the negative eigenvalue spectrum insightful.
>
> ## Strengths and Weaknesses
>
> - The theoretical model of Lyapunov exponents by itself does not provide insights on the training dynamics. Rather, the insights come from checking our experimental results against the theoretical model, leading to the observations discussed, in particular the importance of negative Hessian eigenspectrum (and thus of chaotic training dynamics) to training performance and the continued divergence of network parameters after loss convergence.
>
>
> - While linearized models or neural tangent kernels are an effective tool for analyzing the generalization capability of a model, it is not clear to us how the theory is linked to our analysis of the Lyapunov Exponent. Quadratic models, on the other hand, approximate an ANN with a second order Taylor expansion and thus contain curvature information. But both, linearized and quadratic models, approximate the training process at prior conditions $\theta_0$ to provide insights about training properties, while in the locally chaotic case, we utilize the Hessian at every training step.
> The investigation of the "global" chaos via the mLCE (section 4.2) provides a tool to measure parameter variance at training convergence given similar initial parameters and we suggest further research into a possible connection to generalization properties. We see no equivalence or similarity to the previously mentioned methods, apart from also aiming to help to understand the training process. We are interested to learn about further pre-established connections that appear to be missing.
>
> - The fact that negative Hessian eigenvalues lead to chaotic evolution may not be new to the optimization community, but our impression is that this theoretical connection has not received much attention in the context of neural network training. As we pointed out in the related work section 3, Sasdelli et al. (l. 139) is the only work known to us to have paid attention to it, although they arrive at this connection through modeling of NN training as a time-continuous (rather than discrete) dynamical system and thus miss out on the fact that sufficiently big positive eigenvalues would also lead to chaotic evolution.
> However, positive eigenvalues of this magnitude appear not to be present in the models investigated by us or the community, which is why we argue that positive eigenvalues are not relevant for chaotic evolution of neural networks (section 4.1).
> However, we think you are not wrong in your general assessment that the content in section 2.1 and 2.2 can be seen more as the theoretical background. To better stress our original contribution from this section of the paper, perhaps you would agree that we reformulate contribution 1 as follows:
> > “By modeling ANN training with SGD as a time-discrete dynamical system, we propose a modified SGD algorithm ensuring non-chaotic training dynamics to study the importance of chaos in ANN training”
>
> - Due to the high computational cost involved, we could not extend our analysis to higher-dimensional datasets at this time. However, seeing as we performed our analysis both for an MLP and a CNN architecture, and for SGD with and without momentum, we are confident about the significance of our findings for problems investigated by the Deep Learning community. Furthermore, comparable toy datasets such as MNIST have been used in previous influential work analyzing second order dynamics of NN training (e.g. Sagun et al. 2017).
>
> ## Questions:
>
> - Using Lanczos’ algorithm for the calculation of the eigenvalues was considered and we tried using it in our very first experiments. Indeed, there is an implementation of Lanczos’ algorithm for Pytorch called [Pytorch Hessian Eigenthings](https://github.com/noahgolmant/pytorch-hessian-eigenthings) which we attempted to use, but unfortunately had to discard because it was not fast enough. The custom second-order method we employed enables us to calculate the full spectrum at far greater speed, but at the cost of high memory usage, making it unsuitable for high-dim datasets at the moment, but this could change as VRAM of GPUs increases (for more details, see the O2Grad manuscript we included in supplementary material). We included a comparison titled ```lanczos.ipynb``` with the code.
> Bear in mind that the approach chosen for our experiments in 4.1 and 4.2 requires the highly expensive calculation of the full Hessian eigenspectrum at every single time step. Previous analyses of the Hessian and Lyapunov spectrum in NN training have avoided this problem by investigating only parts of the spectrum (such as investigating on the mLCE directions) or have calculated the full spectrum, but done so only at determined steps during the training (as is the case in paper [1] you referenced).

---

> > ### Comment · Reviewer_tUai · 2022-08-08
> > **response**
> >
> > I've read authors' response. Again, I'm not convinced that the proposed model is not any more useful than local quadratic analysis, so I maintain the score.

---

> > > ### Author Response · Authors · 2022-08-09
> > > **Second Response to Reviewer tUai**
> > >
> > > Thank you for reading our response. We would be thankful if you could pinpoint what aspects of our current analysis you believe to have been shown through quadratic models (ideally by referring us to a paper) so we can more accurately address this issue. To our knowledge, you are referring to Neural Quadratic Models (NQM) from [Quadratic models for understanding neural network dynamics](https://arxiv.org/pdf/2205.11787.pdf). We have taken a careful look at the paper and we think the methods are at best only remotely related and the novelty of our contribution isn’t challenged by this paper.
> > >
> > > Methodically speaking, the approach of the two papers is completely different: The NQM paper approximates the model using 2nd order Taylor series expansion at initial parameters $w_0$ for $t=0$, while we investigate curvature information given parameters $w_t$ at any time step $t$. The authors use this approximation to show that its (loss) evolution is similar to that of neural networks and can therefore be used as a model system. We investigate the importance of training along Hessian eigenvectors, as well as chaos and sensitivity on initial conditions over the period of the whole training. Furthermore, the NQM paper mostly investigates catapult convergence for super-critical learning rates. However, our experiments were conducted for sub-critical learning rates. Thus, there is no direct correspondence between our empirical results and most theoretical results of the NQM paper.
> > >
> > > Going into more detail, we found some aspects where at first glance, the two papers could mistakenly seem to be investigating the same things:
> > >
> > > 1. **Convergence/divergence dynamics**: The NQM paper investigates converging/diverging dynamics exclusively in terms of the loss, but not in terms of model trajectories in phase space, which is the focus of our research. The former does not necessarily imply the latter.
> > >
> > > 2. **Critical learning rate**: The paper and its predecessors use the neural tangent kernel (NTK) to derive a critical learning rate $\eta = 2/\lambda_\max(H)$ at the edge from linear to super-critical training. Using a different chaos-theoretic approach, we derive a similar edge $\lambda_i > 2/\eta$ for the transition to chaotic training dynamics in the (positive) eigenvalues. This connection may arise due to the learning rate $\eta>2/\lambda_\max(H)$ letting the model leave a parabolic area exponentially fast (sensitivity to initial conditions), which in fact could lead to finding a better local minimum which explains the “catapult behavior”. While the connection to our research is interesting, it ties in differently to the research conducted in this paper. Furthermore, the NQM paper says nothing about the importance of negative eigenvalues.
> > >
> > > 3. **Tangent kernel/map**: In the preliminaries, the NQM paper arrives at the critical learning rate through the tangent kernel, which is defined as $\nabla F(w_0) \nabla F(w_0)^T$. This expression looks similar to our finite-time Lyapunov matrix and it seems tempting to think that they are one and the same thing, but they are not. However, the two are related as
> > > $Y^{(t+1,t)} = -\gamma H_{wy}^{(t)} \cdot \nabla F(t)$, where $H_{wy}^{(t)}$ is the Output-Parameter-Hessian at time step $t$ (see the ```tUaiProof.pdf``` for a proof).
> > >
> > > To sum up succinctly where the NQM paper differs from ours in its contributions:
> > >
> > > 1. The NQM paper makes **no predictions on the role of the negative eigenvalues of the Hessian** to the training.
> > > 2. The NQM paper **does not investigate Lyapunov exponents, minimum eigenvalues**, eigenvalue counts, gradient overlaps, among other quantities investigated in our paper.
> > > 3. The paper gives no insight on the **diverging model trajectories** at the end of the training.
> > >
> > > It is possible that the divergence of trajectories and the properties we characterized using Lyapunov exponents can also be derived from a NQM and we think this would be an interesting question for further research. However, as is, the aforementioned paper does not establish this connection or predict most of the behavior empirically investigated in our paper, and we have doubts that the connection is sufficiently trivial to invalidate the novelty of our investigations.
> > >
> > > We have also found another paper using quadratic models to study dynamics of neural networks: [Beyond Linearization: On Quadratic and Higher-Order Approximation of Wide Neural Networks](https://openreview.net/attachment?id=rkllGyBFPH&name=original_pdf). However, we have not found any apparent similarities between their results and ours, which is why we have focused our response on the other paper. Please let us know if we are missing out on something.
> > >
> > > We are happy to conduct a comprehensive technical debate with you on the matter, but we are limited in our ability to address your concerns in the absence of more specific criticism. We would be thankful if you considered updating your score should you find our elaborations convincing.

---

### Official Review · Reviewer_yYv3 · 2022-07-11

**Rating:** 6
**Confidence:** 4
**Soundness:** 3 good
**Presentation:** 3 good
**Contribution:** 3 good

**Summary:**

The authors conduct a thorough hessian-related analysis of training dynamics in neural network training in several medium-sized tasks.  They analyze the eigenspectrum of the hessian throughout training, and investigate the effects of screening descent directions by various criteria related to properties of the hessian.  Additionally, they provide a bevy of theoretical results on their studied dynamics.

**Questions:**

There is a sense in which components of this work are somewhat tautological, and other results might be slightly overstated.  As an example, the authors state:

```
In this paper, we have shown that chaotic dynamics of ANN training by SGD are linked to negative curvature.
```

The authors seem to have _defined_ “presence of chaotic dynamics” to be “local hessian has negative eigenvalues”.  Consider the following alternative: it could be the case that essentially parabolic regions of the loss landscape are stitched together by a small number of hyperbolic descent directions.  Restricting oneself to only the strictly positive eigenvalues then somewhat definitionally traps the performance of a model to within some locally-minimum basin, whereas restricting oneself to the negative-hessian directions is more akin to a kind of truncated BFGS iteration, where descent to the lowest minima is never truly ruled out.

It is not, then, anything particularly “chaotic” about the presence of these negative eigenvalues—only that they’re conduits between more locally quadratic regions.  This would likewise explain why learning trajectories tend to correlate more with the positive curative directions.

Do the authors have evidence that there is something more “chaotic” going on here, other than eigenvalues being negative?

As for the author’s definition of “chaos”:
```
Given a small deviation in the parameter state of a model at any point of the optimization, one should hope to obtain a similar solution at the end of the optimization procedure, and by extension a solution that performs similarly well both on training and validation data. Therefore, we use chaotic to refer to the sensitivity to initial conditions.
```
It might be that the community simply needs a different word to describe what is going on with neural network training.  A model whose training dynamics were purely diffusive would also seem to satisfy the author’s notion of “chaotic”.  I suppose we can identify “sensitive to initial conditions” with “chaotic”, but this might be overloading the word.

Regardless, stepping back for a moment and taking a more thousand foot view, here: it seems that the author’s primary result is that “performance of neural networks degrades when we restrict neural networks to not be able to descend along directions with negative hessian eigenvalues”.  Do the authors agree with this characterization?  Do the authors agree that restricting learning dynamics to positive-semi-definite subspaces of the hessian traps the learning dynamics near a local minimum / saddle point?

**Limitations:**

Beyond the comments above, many of the training curves in the paper seem not to be fully converged, but given the extreme numerical expense of running this analysis, I don’t think it’s the end of the world.

**Strengths And Weaknesses:**

Originality: while studying hessian eigenspectra is not a _new_ idea, the approach considered by the authors does appear novel (i.e., screening of particular directions during training), and has yielded interesting results
Quality/Clarity: The paper is quite high quality and clear
Significance: I expect this paper to be significant, as it gestures at negative components of the hessian being more important than folks otherwise suspected.

---

> ### Author Response · Authors · 2022-08-02
> **Response to Reviewer yYv3**
>
> Thank you for your review and for your positive valuation of our work.
>
> Your assessment is correct that, in the context of this investigation, we focus on a characterization of chaos as “sensitivity on initial conditions” (determined by a Lyapunov exponent > 0) and show that if some Hessian eigenvalue satisfies equation (10) at every time step of the training, then the mLCE of the training is greater than 0 for finite time (Theorem 2.2), so two close trajectories separate exponentially fast. To our knowledge, aside from our focus on “sensitivity to initial conditions”, the definitions chosen follow canonical chaos theory and the special role of negative eigenvalues in our derivations arises naturally. Indeed equation (10) implies that chaotic dynamics (according to our notion) also arise given sufficiently large positive eigenvalues. However, the distribution of eigenvalues in the models investigated by previous works of the community and by us suggest that positive eigenvalues of this magnitude do not occur in “natural” examples.
>
> We agree that the model you propose would explain most of the training dynamics observed and we find this to be in line with our additional analysis done in B.3. Or to put it in your words, “restricting learning dynamics to positive-semi-definite subspaces of the hessian traps the learning dynamics near a local minimum / saddle point”. We currently have no conclusive evidence that there is something more chaotic going on beyond sensitivity to initial conditions. As also discussed with reviewer 3, if the orbits of the trajectories converged against bounded regions in phase space and Lyapunov exponents of 0, this would suggest the presence of proper edge-chaotic behavior (rather than just sensitivity to initial conditions) at the end of the training, but the way we see it, since the trajectories continue to diverge for the time spans investigated even after model performance is fairly converged, it could be that perturbed trajectories do not end up in bounded regions around respective local minima, but that they end up in infinite regions minimizing the loss and keep moving apart along certain axes forever. If this were the case, the behavior at the end of the training would not be edge-chaotic.
> As also commented to reviewer 3, we have made some changes to the manuscript to avoid confusion and an excessive overloading of the word “chaotic”. Please let us know if you think they properly address the issues you stated.

---

> > ### Comment · Reviewer_yYv3 · 2022-08-09
> > **response**
> >
> > Thank you for the modifications and the explanation!

---

### Meta-Review · Area_Chair_JUYn · 2022-08-29

**Recommendation:** Accept
**Confidence:** Less certain

**Metareview:**

This paper studies the role of (local) chaos in determining the training dynamics of neural networks. The authors first introduce a standard global notion of chaos via the Lyapunov matrix and introduce a greedy local version which determines whether the dynamics are “locally chaotic”. The authors relate these dynamics to the eigenvalues of the hessian. The authors close with interesting experiments showing that the chaotic, negative, eigenvalues of the hessian are important for training.

This paper generated some debate and ultimately two reviewers favored acceptance, while one reviewer wanted to reject the paper. The two reviewers who wanted to accept the paper appreciated the new emphasis that the paper brought to the negative eigenvalues of the hessian.  The negative reviewer focused on weak experiments and thought the analysis was too similar to the noisy quadratic model. Ultimately, I do think the analysis done here might provide some insight not present in the NQM, since as the authors note, they consider the spectrum throughout training whereas the NQM considers a noise model on top of a fixed hessian. I share the positive reviewers’ feeling that the results about the negative eigenvalues seem quite interesting. At the same time, I do share the negative reviewer’s issue that the experiments might be too simplistic to draw interesting inferences from.

Ultimately, I do think the pros probably outweigh the cons of accepting this paper. However, I would ask the authors whether the description in terms of chaos makes sense since, as they note, they generally only discuss the local behavior for which the notion of chaos doesn’t necessarily make sense.


**Award:**

No

---

### Decision · Program_Chairs · 2022-09-14

Accept